# Predicting Fecal Indicator Bacteria Using Spatial Stream Network Models in A Mixed-Land-Use Suburban Watershed in New Jersey, USA

**DOI:** 10.3390/ijerph20064743

**Published:** 2023-03-08

**Authors:** Tsung-Ta David Hsu, Danlin Yu, Meiyin Wu

**Affiliations:** 1New Jersey Center for Water Science and Technology, Montclair State University, Montclair, NJ 07043, USA; 2Department of Earth and Environmental Studies, Montclair State University, Montclair, NJ 07043, USA; 3Department of Biology, Montclair State University, Montclair, NJ 07043, USA

**Keywords:** fecal indicator bacteria, land use, microbial water quality modeling, spatial autocorrelation, spatial stream network models, urban and suburban water quality

## Abstract

Good water quality safeguards public health and provides economic benefits through recreational opportunities for people in urban and suburban environments. However, expanding impervious areas and poorly managed sanitary infrastructures result in elevated concentrations of fecal indicator bacteria and waterborne pathogens in adjacent waterways and increased waterborne illness risk. Watershed characteristics, such as urban land, are often associated with impaired microbial water quality. Within the proximity of the New York–New Jersey–Pennsylvania metropolitan area, the Musconetcong River has been listed in the Clean Water Act’s 303 (d) List of Water Quality-Limited Waters due to high concentrations of fecal indicator bacteria (FIB). In this study, we aimed to apply spatial stream network (SSN) models to associate key land use variables with *E. coli* as an FIB in the suburban mixed-land-use Musconetcong River watershed in the northwestern New Jersey. The SSN models explicitly account for spatial autocorrelation in stream networks and have been widely utilized to identify watershed attributes linked to deteriorated water quality indicators. Surface water samples were collected from the five mainstem and six tributary sites along the middle section of the Musconetcong River from May to October 2018. The log_10_ geometric means of *E. coli* concentrations for all sampling dates and during storm events were derived as response variables for the SSN modeling, respectively. A nonspatial model based on an ordinary least square regression and two spatial models based on Euclidean and stream distance were constructed to incorporate four upstream watershed attributes as explanatory variables, including urban, pasture, forest, and wetland. The results indicate that upstream urban land was positively and significantly associated with the log_10_ geometric mean concentrations of *E. coli* for all sampling cases and during storm events, respectively (*p* < 0.05). Prediction of *E. coli* concentrations by SSN models identified potential hot spots prone to water quality deterioration. The results emphasize that anthropogenic sources were the main threats to microbial water quality in the suburban Musconetcong River watershed. The SSN modeling approaches from this study can serve as a novel microbial water quality modeling framework for other watersheds to identify key land use stressors to guide future urban and suburban water quality restoration directions in the USA and beyond.

## 1. Introduction

Good water quality ensures the well-being of urban and suburban residents, as it safeguards public health and provides economic benefits through recreational opportunities [1,2]. However, microbial water quality is often impaired due to expanded impervious areas and the improper management of sanitary infrastructures, such as combined sewer overflows and failing septic tanks, resulting in elevated concentrations of fecal indicator bacteria and waterborne pathogens in urban and suburban waterways [1]. Therefore, it is essential to monitor microbial water quality regularly. Microbial water quality is assessed using fecal indicator bacteria (FIB), such as *E. coli*, fecal coliform, and enterococci. These FIBs originate from the gastrointestinal tract of human or warm-blooded animals and are introduced into the urban and suburban waterbodies through fecal matter [3]. Although FIBs are not pathogens per se, their presence indicates the potential cooccurrence of true pathogens [4]. Water quality criteria were hence established to protect public health based on epidemiological studies that investigate the relationships between the rates of gastrointestinal illness among swimmers and the levels of FIBs [5,6,7,8]. For example, the most recent USEPA’s Recreational Water Quality Criteria uses a geometric mean of 126 CFU of *E. coli* per 100 mL to indicate an estimated illness rate of 36 per 1000 primary contact recreators in fresh water [4]. Exceedance of the water quality threshold indicates an elevated risk of waterborne illness, and the waterbody is deemed impaired and needs to be listed on the List of Impaired Waters (i.e., Clean Water Act 303 (d) List). Water quality restoration plans, such as total maximum daily load (TMDL) plans, are then to be developed to help guide pollution reduction efforts in order to attain water quality standards [9]. Development of a TMDL plan requires identifying and quantifying sources of pollutants and determining the degree of reduction needed to meet the applicable water quality standards [10].

Mathematic models are heavily involved in the process of TMDL development for pollutant load estimation and source allocation in order to identify and quantify the degree of contamination in a waterbody for better remedial plans [11]. Mechanistic models that are based on underlying hydrological and biogeochemical processes, such as streambed bacterial deposition and decay in the natural system, have been widely utilized for TMDL development, such as Soil and Water Assessment Tool (SWAT) and Hydrological Simulation Program-FORTRAN (HSPF) [10,12,13]. These process-based models often demand more expertise in understanding detailed mechanisms and require more time in developing models [11]. On the other hand, empirical models that are based on statistical approaches to estimate pollutant loads or identify associations between stressors and response variables are also widely used in practice [11]. Without needing to first understand the underlying complex processes regarding the fate and transport of contaminants, empirical approaches serve as a statistical alternative to support TMDL development [11]. A variety of regression models have been applied to identify key environmental variables linked to deteriorating water quality indicators, including linear, logistic, and Poisson regressions [14,15,16,17]. Linear regressions are highly accessible, easy to implement and have been widely applied in water quality data analysis. They are broadly accepted among water resource stakeholders [18]. Building a valid regression model, however, often requires a relatively large number of observations and enhanced knowledge to identify the relationships between the response and input variables. While the latter can often be established under a theoretical framework and multivariate and/or nonparametric exploration, acquiring a large number of observations could prove to be difficult, especially when sample collection and processing are costly and take a long time. Oftentimes, compromises have to be made, accepting a larger variance for the sake of a relatively small sample size to proceed with the intended analysis. In addition, linear regression assumes no heteroscedasticity among the residuals and the independence of observations [19]. In reality, since “near things are more related than distant things” [20], observations from adjacent geographic locations tend to have similar characteristics. The term “spatial autocorrelation” or “autocovariance” describes the similarity of measurements as a function of the distance separating them [21,22]. Studies that consider spatial autocorrelation often use Euclidean distance to measure the straight-line distance between measurements and apply it in the terrestrial ecosystems [22]. However, due to the constraints of movement of aquatic organisms and materials within stream channels, for stream water quality modeling, along-channel in-stream distance might be more appropriate [22].

Modeling water quality with stream distance had not previously been well explored until the last decade. Money and Carter [23] developed a predictive model of *E. coli* based on turbidity data with river-based distance. In addition to turbidity, other water quality and quantity parameters and watershed characteristics, such as land use type and locations of point source pollutions, often play essential roles in affecting microbial water quality, and have been widely considered in TMDL implementation plan development [24]. To take into consideration the potential spatial autocorrelation in the modeling process, spatial stream network (SSN) models were developed and applied in recent studies [21,25,26,27,28]. An SSN model employs a pre-established ArcGIS toolset and open source R package to provide an empirical approach to modeling stream water quality in response to watershed attributes with the consideration of spatial autocorrelation to better reflect stream characteristics [29,30]. Autocovariance of the SSN models was constructed based on moving average functions to address spatial autocorrelation and measure the degree of dependence among observations in the stream systems [29]. The conventional autocovariance model assumes isotrophy, indicating that the autocorrelation among observations depends only on the distance separating them, but not on the direction [31]. Moving average functions address the direction-based autocorrelation in the stream systems in tail-up or tail-down models [29,31]. Since SSN modeling relies less on understanding underlying hydrological and biogeochemical processes contributing to water quality but utilizes readily available land use/land cover (LULC) data, it provides greater potential for wider applications by stakeholders engaging in water quality issues in urban and suburban environments with limited resources [22,26]. SSN modeling can also predict FIB at a predetermined interval along the stream channel, providing opportunities to identify hot spots for further water quality monitoring and restoration plans [32].

The Musconetcong River, from which we took water samples for the current study, is located in northwestern New Jersey and forms part of the National Wild and Scenic River System in recognition of its remarkable recreational resources [33,34]. Urban and suburban water recreation provides opportunities to promote physical and mental health among residents, foster environmental stewardship and community engagement, and support tourism and the local economy [2,34]. However, the water quality of the Musconetcong River has been historically impaired due to intensive urbanization and roadway development [33]. In fact, it has been listed in the New Jersey Integrated Water Quality Monitoring and Assessment Report (303 (d) List) due to the high abundance of fecal coliforms in northwestern New Jersey. Historically, human wastewater was not connected to sewer systems in the watershed, and various types of domestic and industrial wastewater effluents were discharged into the river through dysfunctional septic tanks. Other non-point source originated from livestock operations, wildlife, etc., contributing considerably to fecal contaminations throughout this mixed-land-use watershed [35]. In 2003, a TMDL plan was established to address microbial water quality deterioration here, requiring a 93% reduction in fecal coliforms [35]. Since then, various water quality restoration projects have been undertaken, including riparian buffers, prescribed grazing, herd reduction, sinkhole closure, and green infrastructure [33]. In 2018, a water quality monitoring study concluded that there had been a significant improvement in microbial water quality over the past decade; however, further reductions in *E. coli* loads were still required to meet the TMDL goal [33]. Local water quality stakeholders have been developing additional monitoring strategies and water quality restoration plans, with delisting of the Musconetcong River from the 303 (d) List as the goal.

In this study, we aimed to develop empirical models with consideration of the spatial autocorrelation and associated key watershed variables with fecal indicator bacteria in the suburban mixed-land-use Musconetcong River watershed. Specifically, our goal was to analyze patterns of *E. coli* concentrations in response to upstream land use attributes through spatial stream network models. We hypothesize that the elevated *E. coli* concentrations in the Musconetcong River were strongly associated with the upstream urban land use. The results from this study will not only provide insights into identifying key land use stressors to guide future water quality restoration directions in this suburban Musconetcong River watershed, but also serve as a potential water quality modeling framework for different watersheds in the USA and beyond when resources are limited to urban and suburban water quality stakeholders.

## 2. Materials and Methods

### 2.1. Water Quality, Land Use and Precipitation Data

*E. coli* was used as a fecal indicator bacterium as it is more reflective of recent fecal contamination from warm-blooded animals compared to fecal coliforms or total coliforms [4]. Surface water samples were collected twice a month from May to October 2018, with three additional samplings from June to August for a total of 21 events at 5 mainstem and 6 tributary sites located throughout the middle section of the Musconetcong River watershed (Figure 1). No samples were collected when stream channels were dry. The sites were selected based on past water quality monitoring activities showing the significant potential of fecal contaminations. Grab samples were collected aseptically into 1 L sterilized polypropylene bottles and were placed immediately in a cooler during transit. Enumeration of *E. coli* was based on an EPA-approved method using mColiBlue24^®^ Broth (Hach Method 10029). In brief, samples were filtered through mixed cellulose ester membrane filters (0.45 mm, 47 mm, Hach, Loveland Colorado) and then soaked with mColiBlue24^®^ Broth selective medium. The filters were subsequently incubated at 35 °C for 24 h. After incubation, colonies showing a blue/indigo color were recorded as *E. coli*. Tentative *E. coli* colonies were further verified using brilliant green bile (BGB) and lauryl tryptose broth (LTB). The final results were reported as colony forming units (CFU) per 100 mL of water samples. After the processing of the samples, we checked the weather data on the days when samples were collected and determined if certain samples were collected during a storm event based on the weather data. All *E. coli* data were uploaded to the Water Quality Portal (https://www.waterqualitydata.us/, accessed on 5 December 2021).

The 2015 land use/land cover data were acquired from the New Jersey Department of Environmental Protection (NJDEP), Bureau of Geographic Information Systems. A modified Anderson coding system was used to categorize land use data [36]. The total drainage area for the whole Musconetcong River watershed was 120,941.71 acres, with 53.2% forest, 21.2% urban, and 14.0% agricultural land. Four land use categories were extracted as explanatory variables for spatial stream network (SSN) modeling, including urban (1000), pasture (2120), forest (4000) and wetland (6000). Due to the sampling cost and length, the sample size in the current study was not particularly large (21 events). While more detailed land use land cover products, such as the USGS’s Multi-Resolution Land Characteristics (MRLC) land cover, are available, to avoid losing too many degrees of freedom in the empirical study, we elected to use the relatively coarse land use land cover categorization. Urban land use classification covered land use characteristics from rural single units to high-density multiple residential dwellings, commercial buildings, and industrial areas. Houses with various dwelling units or a high density in the study area were all connected to septic tanks instead of sewage pipes for waste disposal (Personal Communication with the Musconetcong Watershed Association). The study’s purpose was to investigate in general how urbanization within a watershed impacts water quality. The pasture subgroup was selected from the agricultural land use categories (2000), as failing septic systems and poor livestock management practices were previously listed as major threats to microbial water quality [35]. Precipitation data were obtained from NJDEP, the Department of Water Monitoring & Standards (https://njdep.rutgers.edu/rainfall/, accessed on 5 December 2021). The rainfall threshold for a storm event was defined as 12.7 mm (0.5 inch) within 36 h, the same definition used in the Watershed Restoration and Protection Plan for the Musconetcong River Watershed created by the Rutgers Cooperative Extension Water Resources Program [37]. A summary of analytical methods and data sources used in this study is shown in Table 1.

### 2.2. Spatial Stream Network Modeling

To explore the impact of land use in the watershed on water quality, we extended from traditional regression analysis to adopt the spatial stream network model in this study [22,38,39]. Spatial stream network models are generalized linear mixed models allowing explanation of variance in observations with both fixed and random effects due to spatial autocorrelation among geographically observed events [40,41], which can be expressed as follows:y = Xβ + z + ε
where y is a vector of observations, X is a matrix for fixed effects that explain the variance of the observations that can be captured by general spatial patterns, β is a vector of coefficients for X, z accounts for random effects resulting from spatial autocorrelation that cannot be explained by fixed effects, and ε is a vector of independent random errors [21,32]. The spatial autocorrelation structures that describe the random effects can be modeled with linear with sill, Mariah, exponential, Epanechnikov, and spherical models [29]. Parameters for spatial autocovariance functions include nugget, partial sill, and range. The nugget of the autocovariance function describes the portion of variance that cannot be explained, the partial sill accounts for the variance that depends on the distance among observations, and the range indicates the minimal distance at which observations are no longer spatially correlated [32]. SSN models can also incorporate autocovariance models based on Euclidean distance, accounting for terrestrial and atmospheric factors that are stream-independent [22]. Both tail-up and tail-down autocovariance structures were established to estimate spatial autocorrelation. In tail-up models, moving average functions run upstream only and split at confluences. Spatial weights are required to estimate the proportion of upstream tributary influence based on flow volume, watershed area, or other relevant attributes [22]. Tail-up models are only applicable to flow-connected sites, where water directly flows from an upstream to a downstream site characterized by passive downstream movement, e.g., temperature, bacteria, or sediment [22]. In tail-down models, moving average functions run in the downstream direction unilaterally until reaching the most downstream location of the stream network, indicating the possibility of autocorrelation among all locations [29]. In addition to flow-connected sites, tail-down models can be applied to flow-unconnected sites, where some upstream movement is expected to facilitate the connectivity for a given attribute, e.g., fish or macroinvertebrates [22].

In order to perform spatial stream network (SSN) modeling, an SSN object was first created using STARS geoprocessing toolset [21] in ArcGIS 10.8 (ESRI, Redlands, CA, USA). STARS is used to build a landscape network (LSN) as a personal geodatabase to represent spatial relationships, such as flow connectivity, direction, and distance [29] and transform the geometry, attribute data, and topological relationships among features of GIS datasets into an SSN object that can be easily accessed and analyzed in R statistical software with the ssn package [30] for SSN modeling. The process included building a landscape network (LSN), creating reach contributing areas [42], calculating RCA attributes and area, accumulating watershed attributes, incorporating the sampling locations into the LSN, calculating watershed attributes, and calculating upstream distance, segment proportion influence, and additive function [43]. In the watershed of the current study, we obtained a total of 427 nodes in the stream network. The entire process of generating the SSN object was automated, and once generated in ArcGIS, the SSN object was then exported to R statistical software [44] for SSN modeling.

In this study, the ssn package in R [30,39] was used to fit spatial stream network models to the observation data based on 11 sampling locations. Two models were established and calibrated. The first model used the geometric mean concentrations of *E. coli* for all sampling events as the response variable; the second model used the geometric mean concentrations of *E. coli* during storm events only as the response variable. Shapiro–Wilk tests were used to test the normality of the distributions of the original values [45]. If the normality assumption was not satisfied, the values of geometric mean concentrations were log-transformed. A nonspatial model based on an ordinary least squares regression and two spatial models based on Euclidean and stream distance, respectively, were established to incorporate four upstream watershed attributes as explanatory variables, including urban, pasture, forest, and wetland. Percentage upstream land use values were derived from dividing the total accumulated area of upstream land use by the total upstream catchment area processed and generated via the STARS toolset. Variance inflation factors (VIF) were calculated to check multicollinearity among the four land use variables. Tail-up models were applied in the spatial autocovariance functions for the SSN modeling based on stream distance, as the movement of bacteria along the stream networks is characterized by passive downstream transport [22]. Likelihood approaches were utilized to estimate the regression coefficients b for the matrix of fixed effects [46]. Maximum likelihood (ML) estimators have been shown to be greatly biased when the number of observations is small [47], and thus restricted maximum likelihood (REML) was used. An alpha value of 0.05 was used to evaluate whether the relationship between watershed variables (urban, agricultural, forest, or wetland) and *E. coli* concentrations in the models was statistically significant. The root mean square prediction error (RMSPE) of the leave one out cross validation (LOOCV) was used as the criterion to compare the predictive capability among models. LOOCV was performed by excluding one observation at a time. Submodels were calculated and compared with the excluded samples [48]. A lower RMSPE of the LOOCV indicated a better model performance [32]. The coefficient of determination (R^2^) was used to measure the proportion of variance in the observations explained by the fixed or random effects of each model.

### 2.3. Predicting E. coli Concentrations

One of the practical uses for establishing the empirical SSN model for the Musconetcong River watershed is that we can use the model to predict the concentration of *E. coli* along the river. Predictions were performed for the geometric mean concentrations of *E. coli* for all sampling events and during storm events based on Euclidean distance and stream distance models, respectively. A total of 816 predictive sites were assigned evenly across the Musconetcong River stream network at a 300-m interval. Determination of predictive sites was conducted using the STARS geoprocessing toolset [21] in ArcGIS 10.8. Predictions of *E. coli* concentrations were performed in the R statistical software [44] with SSN package [30]. The results were illustrated on stream maps illustrating both the log_10_ geometric mean concentrations of *E. coli*. The capability of predicting the concentration of *E. coli* provides both a means to assess modeling performance and a supplement to the regular sampling routine for water quality monitoring and management.

## 3. Results

### 3.1. Microbial Water Quality and Land Use

All 21 sampling events were recorded for most of the locations, except for T1 (3) and T3 (19). Among them, three were defined as storm events (accumulated rainfall within 36 h was 13.2, 22.9, and 41.9 mm, respectively), except for T1 (one event, accumulated rainfall was 22.9 mm). The geometric mean concentrations of *E. coli* for all sampling events among the mainstem sites were similar (240 to 296 CFU/100 mL), while those among the tributary sites had a greater variation (73.3 to 1355.6 CFU/100 mL). Four of the six tributary sites exceeded the geometric mean threshold of *E. coli* (126 CFU/100 mL) compared to one of the five mainstem sites for all sampling events (Figure 2). During storm events, the geometric mean concentrations of *E. coli* among the mainstem sites ranged from 303 to 811 CFU/100 mL), while those among the tributary sites also had a greater variation (585.7 to 7300 CFU/100 mL). All of the sites exceeded the geometric mean threshold of *E. coli* (Figure 2). The upstream land use characteristics of sampling locations were similar among the mainstream sites, while those of tributary sites exhibited a greater variety as each site demonstrated unique patterns in this suburban mixed-land-use watershed (Figure 3). The percentage of upstream urban land ranged from 21.3% to 22.5% and from 8.2% to 57.4% for the mainstem and tributary sites, respectively. The percentage of upstream pasture varied from 0.0% to 0.2% and 0.1% to 20.3% for the mainstem and tributary sites, respectively. The percentage of upstream forest ranged from 41.0% to 42.9% and from 14.6% to 58.7% for the mainstem and tributary sites, respectively. The percentage of upstream wetland ranged from 6.3% to 6.7% and 0.3% to 6.4% for the mainstem and tributary sites, respectively. Overall, pasture land dominated the study area, with urban lands clustered toward the upper portion. Forest land is concentrated toward the upstream area of tributaries in the study area (Figure 1).

### 3.2. Spatial Stream Network Modeling

Three individual spatial stream network (SSN) models with three distinct autocovariance structures were constructed for each response variable in this study, including no spatial correlation (ordinary least squares), straight-line distance (Euclidean distance), and along-channel in-stream distance (stream distance). Variance inflation factors (VIF) for all four land use variables were below 2, indicating that no multicollinearity among them was detected. The original values of the geometric mean concentrations of *E. coli* were not normally distributed. Therefore, the original values were log-transformed to satisfy the normal distribution assumption. Table 2 shows the SSN model autocovariance structures, coefficients for explanatory variables, and performance. Applying SSN models identified that upstream urban land was positively and significantly associated with the log_10_ geometric mean concentrations of *E. coli* for all events and during storm events, respectively (*p* < 0.05). Upstream pasture land was positively and significantly correlated with the log_10_ geometric mean concentrations of *E. coli* during storm events only (*p* < 0.05). Although insignificantly, the geometric mean concentrations of *E. coli* for all sampling events demonstrated negative correlations with upstream wetland (*p* = 0.06). SSN models improved the overall coefficient of determination (R^2^) as the additional portion of variance can be explained by the random effects attributed to the spatial autocorrelation. In fact, based on stream distance, the overall SSN models explained nearly 100% of variance compared to the ordinary least squares models, with 20% of variance being unaccounted for. The lowest level of unexplained variance was found for SSN models based on stream distance for both response variables (Nugget < 0.0001). Prediction errors also improved from non-spatial to spatial models. SSN models based on Euclidean distance showed the lowest value of LOOCV RMSPE for log_10_ geometric mean concentrations of *E. coli* for all sampling events (1.110), while the lowest value of LOOCV RMSPE was seen for log_10_ geometric mean concentrations of *E. coli* during storm events modeled based on stream distance (1.290). Overall, the distance with no spatial autocorrelation observed for log_10_ geometric mean concentrations of *E. coli* for all sampling events was shorter than that for the storm events based on either Euclidean (Range 2.31 vs. 3.11 km) or stream distance (Range 2.87 vs. 3.37). The best stream distance autocovariance functions were linear with sill tail-up for both log_10_ geometric mean concentrations of *E. coli* for all sampling events and during storm events.

### 3.3. Predicting E. coli Concentrations

Euclidean distance and stream distance models were chosen to predict the log_10_ geometric means concentrations of *E. coli* for all events and during storm events, respectively, due to lower root mean square percentage error (RMSPE) values (Table 2). Figure 4 shows both the geometric mean concentrations of *E. coli* of the 11 sampling locations (stars) as well as the predicted geometric mean concentrations of *E. coli* (solid circles) in the study area. Overall, the predicted geometric mean concentrations of *E. coli* were higher during storm events (Figure 4b) than for all sampling events (Figure 4a). Predictive values for the geometric mean concentrations of *E. coli* ranged from 1 to 5012 CFU/100 mL and from 43 to 30,903 CFU/100 mL for all events and during storm events, respectively.

## 4. Discussion

### 4.1. Spatial Stream Network Modeling Performance

Spatial stream network modeling provides an empirical approach for urban and suburban water quality stakeholders to analyze the spatial distribution of parameters without the need to first understand the underlying hydrological and biogeochemical processes that lead to water quality impairment. The application of SSN models has been introduced to a wide range of water quality indicators, such as temperature [49], dissolved oxygen [50], total phosphorus [26], and macroinvertebrates [51,52]. However, the modeling of fecal indicator bacteria using SSN models had not been documented until recently [32,53]. Holcomb and Messier [53] modeled fecal coliforms in a mixed-land-use watershed in North Carolina and identified agricultural land use, forest cover, antecedent precipitation, and temperature as being strongly associated with mean fecal coliform concentrations. Neill and Tetzlaff [32] used *E. coli* as a fecal indicator bacterium in an agriculture-dominant watershed with sporadic urban development in Scotland and found that the anthropogenic impact index (lumped indicator for potential contamination from human point sources) was significantly correlated with 5th, 50th, and 95th percentile *E. coli* concentrations. Similar to Neill and others (2018), modeling based on stream distance in this study demonstrated improvement in the total R square and root mean square percentage error (RMSPE) for geometric mean concentrations of *E. coli* during storm events. However, only an improvement in the total R square was observed for the model based on stream distance for geometric mean concentrations of *E. coli* for all sampling events. This may be due to limitations regarding the number of sites as well as the locations of the mainstem sites, as they were relatively highly clustered compared to the overall watershed. An improvement in RMSPE was not observed in Holcomb et al. (2018) either. In this study, the best autocovariance functions were linear with sill with stream distance for both log_10_ geometric mean concentrations of *E. coli* for all sampling events and during storm events, indicating a pattern in which the variability among observations increases linearly with the separation distance until it reaches the maximum difference [54]. Overall, the range for the SSN models for storm events was greater than that for all sampling events, and the stream distance models had a greater range than the Euclidean distance models did, indicating heavier impacts on *E. coli* concentrations from upstream areas during storm events.

### 4.2. Suburban Land Use and Microbial Water Quality

Various factors could lead to the deterioration of microbial water quality in urban and suburban waterways [1]. In this study, upstream urban land was identified as being positively correlated with either log_10_ geometric mean concentrations of *E. coli* for all sampling events or during storm events only. Using the same SSN modeling approach, Neill and Tetzlaff [32] identified human (leaking sewage pipes and failing septic tanks) and farmyard sources as being significantly associated with 5th, 50th, and 95th percentile concentrations of *E. coli*. A variety of mechanisms, such as failing onsite wastewater treatment systems (i.e., septic tanks), combined sewer overflows (CSOs), sanitary sewer overflows (SSOs) and urban runoffs can introduce human fecal contamination into adjacent waterways in urban and suburban watersheds [1]. Increased risks of childhood emergency department visits and infectious diarrhea associated with combined sewer overflow and septic sites have been documented [55,56]. In addition to urban land use, upstream pasture land was found to be positively and significantly associated with log_10_ geometric mean concentrations of *E. coli* during storm events in this study. Holcomb et al. (2018) also reported similar significant relationships between agricultural land use and fecal coliform. Ill-considered agricultural practices in suburban watersheds, such as concentrated animal feeding operations, overgrazing and manure applications could lead to water quality deterioration [57,58]. Davies-Colley et al. (2004) also demonstrated that a dairy cow herd produced more than 50,000 CFU/100 mL of *E. coli* when crossing a stream. A significant relationship was found between upstream pastureland use and *E. coli* for storm events only in this study, illustrating the impact of agricultural runoff on microbial water quality during storm events. For instance, higher concentrations of *E. coli* and *Salmonella* were identified in irrigation ponds at two produce farms after rain events [59]. Noteworthily, a negative association (although only marginally significant, *p* = 0.06) was identified between upstream wetlands and geometric mean concentrations of *E. coli* for all sampling events. Wetlands can provide essential ecosystem services, such as water purification and runoff reductions [60]. Reductions in fecal indicator bacteria or pathogens provided by wetlands have been well documented [61,62,63]. Hsu et al. (2017) reported an average 22.3% reduction in *E. coli* across two wetlands receiving inflow from urban waterways [63]. The results from this study also reinforce the concept of using wetlands as the best management practice for microbial water quality restoration. For example, a variety of constructed wetlands have been implemented to treat domestic and agricultural wastewater as well as urban stormwater runoff [64].

### 4.3. Extreme Weather Conditions and Microbial Water Quality

All of the geometric mean concentrations of *E. coli* during storm events exceeded the geometric mean threshold of *E. coli* of 126 CFU/100 mL in this study. Extreme weather events, such as heavy precipitation, have been well documented to degrade water quality by elevating the concentrations of FIB in urban and suburban waterways through stormwater runoff and combined and sanitary overflows [1]. These heavy rainfall events significantly contributed to waterborne disease outbreaks in the United States and Canada [65,66]. In the United States, among the 548 reported outbreaks analyzed from 1948 through 1994, 68% of them were preceded by the highest 20% of precipitation events [65]. In Canada, the heaviest 7% of precipitation events increased the relative odds of an outbreak by 2.3-fold [66]. In fact, more than 400,000 cases of acute gastrointestinal illness (AGI) were attributed to a drinking water treatment plant overwhelmed by high turbidity load after a period of heavy precipitation in Milwaukee, Wisconsin in 1993 [67]. Downstream of the Musconetcong River in Philadelphia, where the Delaware River serves as the drinking water source, a significant increase in waterborne AGI following precipitation above the 95th percentile was documented [68]. Under the current trend of global climate change, the National Climate Assessment projected an almost 50% increase in the total annual precipitation falling in the heaviest cases (1%) by the late 21st century under the higher scenario (RCP 8.5) in the northeastern United States [69]. This will further increase the frequency and intensity of urban stormwater runoffs and combined and sanitary overflows, degrading microbial water quality and increasing potential public health risk in the Musconetcong River and downstream drainage area in the future if no additional water quality protection action is implemented. The land use variables identified from the SSN modeling results can be taken into consideration when developing water quality restoration or climate resiliency plans to address water quality and public health concerns in urban and suburban areas, such as establishing municipal separate storm sewer system (MS4), expanding public sanitary sewer systems, regulating private sewage disposal practice, and promoting the use of green infrastructure.

### 4.4. Predicting Fecal Contamination Hot Spots and Future Directions

Since field water sampling is time-consuming and costly, water quality modeling can serve as a cost-effective and timely approach to enhance existing monitoring programs in conjunction with actual field samples. It can also provide stakeholders with a mean to quickly identify “hot spots” during the site selection process for further water monitoring and water quality enhancement actions. Spatial stream network predictive modeling provides a feasible approach to predict riverine *E. coli* concentrations with a predetermined stream-distance interval. The predicted results can be used to prioritize sites for further monitoring and subsequent restoration practices. Combined with local knowledge, the modeling results can be tailored to form a localized site-specific management approach. In this study, predictive sites near T1 in a catchment basin with a high percentage of urban land showed elevated *E. coli* concentrations for all sampling events and during storm events. Another example is shown in the next four predictive sites downstream of T4. The predicted value for *E. coli* for all sampling events was below the geometric mean threshold (126 CFU/100 mL). However, the predicted value for storm events was above the single-sample maximum threshold (235 CFU/100 mL), indicating the significant impact of agricultural runoff as a result of heavy precipitation on microbial water quality. Different localized water quality restoration approaches should be developed for both areas, such as improving the maintenance of septic systems in urbanized areas and establishing riparian buffers in agriculture-dominant sites.

For TMDL process development, SSN models become a cost-effective approach along with field sampling programs for water quality stakeholders to engage in watershed management with the support of readily available land use land cover data, open-source R packages, and geographic information systems. In fact, the current study area was located in one of the TMDL segments of the Musconetcong River, and additional segments are located both upstream and downstream of the current study area [35]. Therefore, the same SSN approach can be extended to additional TMDL segments within the Musconetcong River watershed to provide a comprehensive understanding for integrated water quality management (NJDEP, 2003). Incorporating additional sampling locations is essential to encompass a larger watershed to improve SSN modeling. Money and Carter [23] suggested “a minimum of 10–50 data points should exist to construct a correlation model depending on the watershed size”. In our current study, because of cost and the size of our watershed, we only collected 21 data points to establish the model. The sample size is on the small end and does not have sufficient degrees of freedom for us to set up a model training and testing routine. While water monitoring is an ongoing endeavor for environmental management teams at both the local and state levels, we hope in the future that expanding sampling locations will allow model validation to encompass both training and validation datasets. It was also suggested that the placement of sampling locations in relation to the whole watershed could impact the outcome of the spatial modeling [38]. For instance, a better overall error map may be produced by placing sites close to the origins of tributaries, while a good estimate of tail-up autocovariance functions could be generated by selecting sites between confluences. In addition, a spatial stream predictive model in conjunction with other field and laboratory observations (e.g., sanitary survey or microbial source tracking) can identify sites with greater contamination potential and sources of fecal contamination. This is essential for a thorough watershed management plan to address impairment and TMDL source reductions.

Although beyond the scope of SSN modeling, temporal variations also affect the outcome when modeling microbial water quality. Holcomb and Messier [53] included a time component in their geostatistical models and obtained improved prediction errors than models considering spatial effects only. The effects of precipitation patterns could also be investigated when the temporal effects are incorporated into spatial modeling to provide a comprehensive understanding of microbial water quality dynamics in response to a variety of environmental variables and weather scenarios. Overall, the current study provides critical insights into assessing the amount of fecal contamination to guide further monitoring and management activities and serves as potential water quality modeling framework for urban and suburban watersheds in the USA and beyond.

## 5. Conclusions

In this study, we successfully applied spatial stream networks (SSN) to model elevated concentrations of *E. coli* in the suburban mixed-land-use Musconetcong River watershed in response to upstream land use attributes, including urban, pasture, forest, and wetland. The SSN model is essentially a spatial statistical model designed for modeling the statistical relationships between variables on stream networks [23]. The modeling results suggest that upstream urban land was positively and significantly associated with log_10_ geometric mean concentrations of *E. coli* for all sampling events and during storm events, respectively. Upstream pasture land was positively and significantly correlated with log_10_ geometric mean concentrations of *E. coli* during storm events only. Although only marginally significantly, the log_10_ geometric mean concentrations of *E. coli* for all sampling events demonstrated a negative relationship with upstream wetland land use as per the SSN model calibration. Applying SSN modeling based on spatial distance also demonstrated improved model performance over non-spatial models. The SSN modeling results concur with previous findings that anthropogenic sources represent the main threat to microbial water quality in the Musconetcong River watershed. The prediction of *E. coli* concentrations by SSN models identified potential hot spots prone to water quality deteriorations. Future directions could benefit from incorporating additional sampling locations beyond the current section of the watershed and introducing temporal effects in the spatial modeling. Nevertheless, with the support of publicly available watershed attribute data, a pre-established ArcGIS toolset and the open-source R package, we believe that SSN models can be employed by other urban and suburban water quality stakeholders to assist in their water quality monitoring and restoration needs when the resources are constrained.

## Figures and Tables

**Figure 1 ijerph-20-04743-f001:**
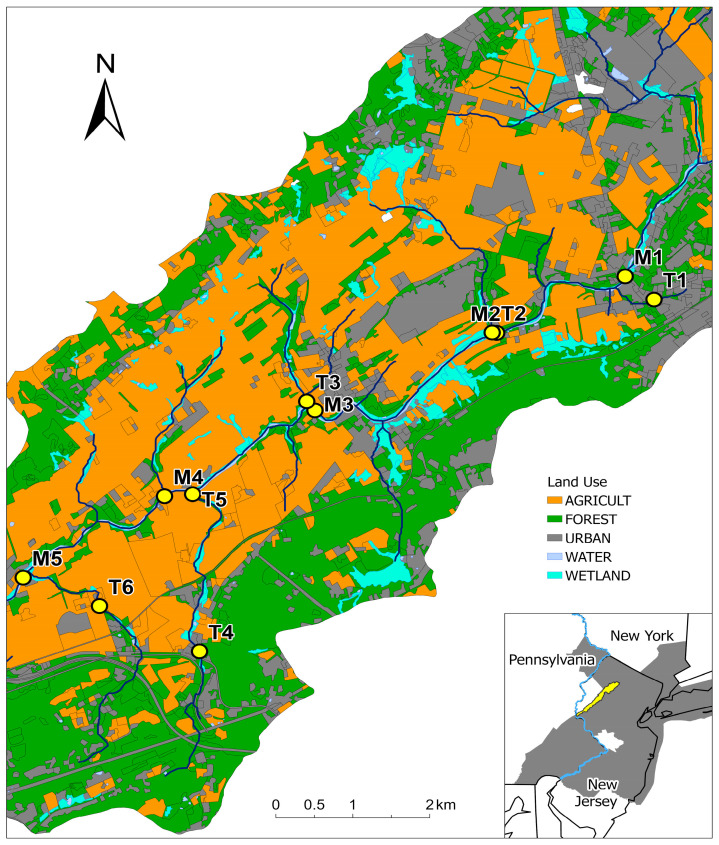
Sampling locations and adjacent land use types in the Musconetcong River watershed in this study. Sampling locations are denoted by yellow dots. The gray area on the map inlet indicates the New York–New Jersey–Pennsylvania metropolitan area (Source: United State Census Bureau).

**Figure 2 ijerph-20-04743-f002:**
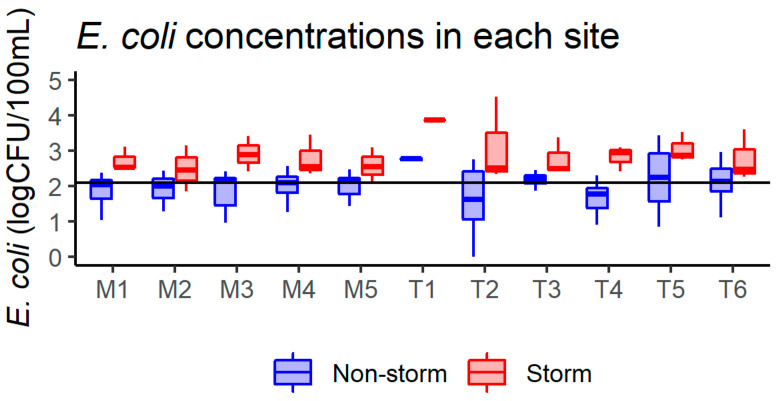
*E. coli* concentrations in a logarithmic scale for each site in this study. Red boxes indicate range of *E. coli* concentrations during storm events. Blue boxes indicate range of *E. coli* concentrations during non-storm events. The threshold of the rainfall amount for a storm event is 12.7 mm (0.5 inch) within 36 h. Black line denotes the geometric mean threshold of *E. coli* concentration of 126 CFU/100 mL (2.1 log10 CFU/100 mL). M indicates mainstem sites, whereas T indicates tributary sites of the sampling locations.

**Figure 3 ijerph-20-04743-f003:**
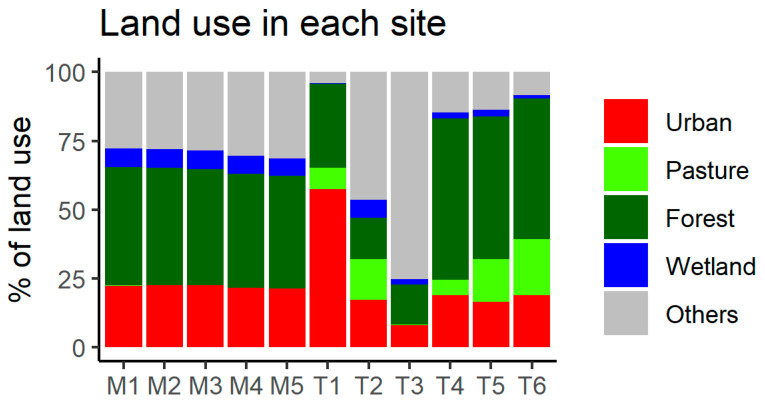
Percentage of each land use for each site in this study. Only the land use classifications used for the spatial stream network (SSN) models in this study are shown, including urban (red), pasture (light green), forest (dark green), and wetlands (blue). M indicates mainstem sites, whereas T indicates tributary sites of the sampling locations.

**Figure 4 ijerph-20-04743-f004:**
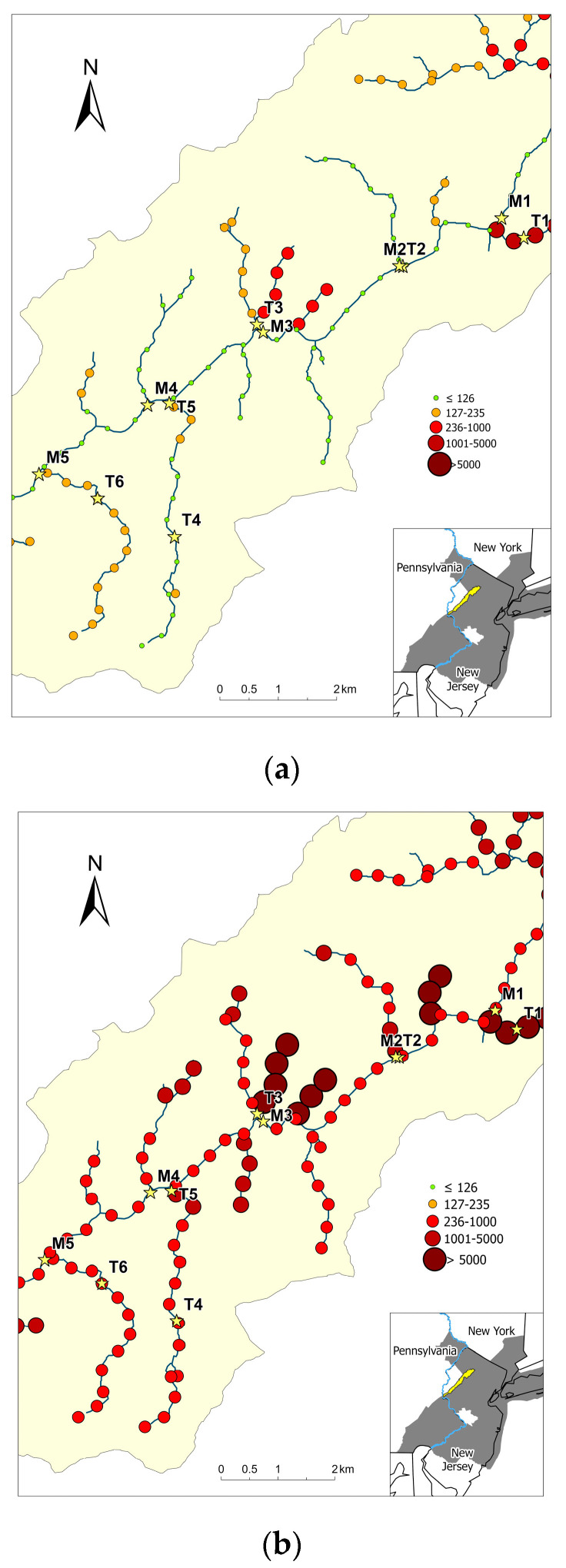
Prediction of (**a**) overall geometric mean concentrations of *E. coli* and (**b**) geometric mean concentrations of *E. coli* during storm events. Color and size of the circles denote the geometric mean concentrations of *E. coli*, with larger and darker circles indicating greater geometric mean concentrations of *E. coli*. Stars indicates the sampling locations in this study.

**Table 1 ijerph-20-04743-t001:** Analytical methods and data sources used in this study.

Data	Sources/Methods
*E. coli*	EPA approved Hach mColiBlue24^®^
Urban land use	NJDEP, Bureau of Geographic Information Systems
Pasture land use	NJDEP, Bureau of Geographic Information Systems
Forest land use	NJDEP, Bureau of Geographic Information Systems
Wetland land use	NJDEP, Bureau of Geographic Information Systems
Precipitation	NJDEP, Department of Water Monitoring & Standards

**Table 2 ijerph-20-04743-t002:** SSN model autocovariance structures, coefficients for explanatory variables, and performance for two response variables (overall geometric mean of *E. coli* and geometric mean of *E. coli* during storm events.

Response Variable	Autocovariance					Coefficients for Explanatory Variables				Model Performance			
	Model	Structure	Nugget	Partial Sill	Range (km)	% Urban	% Pasture	% Forest	% Wetland	Fixed Effect R^2^	Random Effect R^2^	Total R^2^	LOOCV RMSPE
log_10_ Geomean (overall)	OLS	Nugget only	0.041	NA	NA	** 0.018*	−0.003	−0.006	*^ −0.066*	0.81	NA	0.81	1.234
	Euclidean Distance	Gaussian Euclidean + Nugget	0.028	0.014	2.31	** 0.018*	−0.003	−0.005	*^ −0.068*	0.83	0.06	0.89	1.110
	Stream Distance	LinearSill Tailup + Nugget	0.000	0.040	2.87	* 0.018	−0.003	−0.006	*^ −0.068*	0.81	0.19	1.00	1.226
log_10_ Geomean (storm)	OLS	Nugget only	0.054	NA	NA	** 0.021*	0.014	−0.008	−0.026	0.83	NA	0.83	1.396
	Euclidean Distance	Gaussian Euclidean + Nugget	0.033	0.005	3.11	** 0.022*	0.016	−0.008	−0.022	0.84	0.02	0.86	1.346
	Stream Distance	LinearSill Tailup + Nugget	0.000	0.038	3.37	** 0.021*	** 0.019*	−0.009	−0.024	0.83	0.17	1.00	1.290

* *p* < 0.05, ^ *p* = 0.06.

## Data Availability

Not applicable.

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
