# Peer review of "Predicting Fecal Indicator Bacteria Using Spatial Stream Network Models in A Mixed-Land-Use Suburban Watershed in New Jersey, USA"

_ijerph, 2023, doi:10.3390/ijerph20064743_

Round 1

Reviewer 1 Report

This study examines the use of SSN models to predict E. coli in an impaired watershed in New Jersey. The authors report E. coli concentrations are correlated with upstream land use in the watersheds and show SSN models can provide a very good fit to observed data (high R^2), indicating the models may be appropriate for predictive applications. Overall, the study appears sound, although the small sample size is somewhat concerning.  The manuscript can be improved by incorporating the following suggestions.

Section 1. The introduction is quite long. Consider trimming where possible.

Section 2.1: Some basic information about the study watershed is needed. What is the overall land cover makeup (i.e., percent developed, forested, etc.).

-What is the drainage area of the watershed? I do not see this mentioned anywhere in the manuscript.

-Does the analysis of E. coli follow a standard method or EPA method? If so, cite the analysis method you are using.

Lines 179: Do you have any information about the location/abundance of septic tanks that could be included in the discussion? It would be interesting to see if areas of high septic tank density were correlated with high E. coli.

Figure 1. Suggest including the stream/channel network on this map and the drainage area boundaries. The inset map is not helpful to international readers that are not familiar with this area. The scale needs to be increased to provide a wider perspective.

Line 171. Why was this land cover source used? This data is very coarse - lumped all “urban” land uses together, which does not enable you to identify specific urban sources. What is the spatial resolution of this data? The MRLC provided land cover data for 2016 and 2019 that breaks down developed areas into open space, low, median and high density development. This might be a better land use source- https://www.mrlc.gov/data

Line 182. How much wastewater is being discharged and where are the plants located. This information should be available from NPDES permits and compliance monitoring reports. Could this impact your results?

Section 2.2:  It would be good to include a table that lists all the input variables/data used and the source/method by which they were obtained. I understand they are listed in the text but summarizing them in a table would be helpful.

Lines 227-227. This sentence is awkward – please reword. “The response variable were geometric mean concentrations of E. coli for all sampling dates and during storm events, respectively.”

Section 3.1. How many events were considered storm vs. non-storm events? I suggest including the number of events on the boxplot.

Line 282-284: Please reword this sentence as it is confusing. It leads the reader to believe that three models were used considering no spatial correlation.

Line 300: Change “wetland use” to “wetland land use”

Figure 4: “Size of the circles indicate the standard errors of the predictions, with larger ones showing smaller standard errors.”  Is this correct? Why use larger markers for smaller standard errors?

Maybe you need to change the size range of the circles. It is difficult to see any variation in size at this scale.

I also suggest using a different color scheme - the stream layer tends to blend in with the points, especially for fig 4a.

I don’t think the x and y coordinate information is important here. I suggest a background of a watershed map.

Section 3.3: It would be interesting to see a plot of observed vs. predicted E. coli with a 1:1 line. This would illustrate the model fit across the range of observations – could be color coded as storm/non-storm.

I suggest referring to “study sites” as “sampling locations.” It is just one study site (the Musconetcong River watershed).

Section 4.3. Were the rainfall events observed during the sampling period considered extreme events? I do not see any information provided about the storm depth/intensity.

Line 454: I strongly disagree with this statement “Since field water samplings are time-consuming and costly, water quality modeling can serve as a cost-effective and timely alternative.” Models should only be used in conjunction with sampling programs to enhance our understanding. I would not suggest modeling as an alternative to sampling.  

Line 458: Again, characterizing modeling as an “alternative” to sampling is very problematic. It can be used to enhance the information we can gain from sampling but never as an alternative.

Line 477-480: Again, I would not characterize probabilistic modeling as an alternative to sampling. It can increase the information we are able to gain from sampling, but it is NOT an alternative.

Line 489: “at least 10 – 50 observations” This is confusing. It is 10 or 50. That is a very large range. Ten samples seem very low to establish a robust spatial autocovariance model.

Section 4.4. I think you need to add some discussion about how increased sampling would allow for model validation. You do not include model validation in this study, but increased sampling would allow for this. Ideally, you would have had a dataset large enough to separate into training and validation datasets for your model.

Line 521: “all and during storm events.” This wording is awkward, please revise.

Line 526: Change “wetland use” to “wetland land use”

Author Response

This study examines the use of SSN models to predict E. coli in an impaired watershed in New Jersey. The authors report E. coli concentrations are correlated with upstream land use in the watersheds and show SSN models can provide a very good fit to observed data (high R^2), indicating the models may be appropriate for predictive applications. Overall, the study appears sound, although the small sample size is somewhat concerning.  The manuscript can be improved by incorporating the following suggestions.

Response: Thank you for your critical comments. Due to constraints in field work and sample survey, we strive to provide the best possible sample size under these constraints. We hope to gain your understanding and will strive address your concerns to the fullest.

Section 1. The introduction is quite long. Consider trimming where possible.

Response: Thank you for your suggestion. Following your suggestion, we have trimmed the introduction section with only relevant information retained.

Section 2.1: Some basic information about the study watershed is needed. What is the overall land cover makeup (i.e., percent developed, forested, etc.).

Response: Thank you for your meticulous review. Relevant information was added.

-What is the drainage area of the watershed? I do not see this mentioned anywhere in the manuscript.

Response: Thank you again. We have added the relevant information in the revision.

-Does the analysis of E. coli follow a standard method or EPA method? If so, cite the analysis method you are using.

Response: We followed the EPA approved Hach method. Details are added in the Section 2.1.

Lines 179: Do you have any information about the location/abundance of septic tanks that could be included in the discussion? It would be interesting to see if areas of high septic tank density were correlated with high E. coli.

Response: Thank you for your insightful suggestion. We agree with you and had the same idea when conducting data analysis, but unfortunately, the watershed group we’ve worked with did not have the specific information about the location/abundance of septic tanks either. We have included that in the discussion too.

Figure 1. Suggest including the stream/channel network on this map and the drainage area boundaries. The inset map is not helpful to international readers that are not familiar with this area. The scale needs to be increased to provide a wider perspective.

Response: Thank you for your suggestion. Following your suggestion, we have modified the original map to include stream network and watershed boundaries. The scale of the map inlet, however, remains the same to highlight the proximity of the study area to the New York-New Jersey-Philadelphia metropolitan. To provide better geographic reference, the name of the states has been spelled out instead of using abbreviation. We hope the modification could gain your support.

Line 171. Why was this land cover source used? This data is very coarse - lumped all “urban” land uses together, which does not enable you to identify specific urban sources. What is the spatial resolution of this data? The MRLC provided land cover data for 2016 and 2019 that breaks down developed areas into open space, low, median and high density development. This might be a better land use source- https://www.mrlc.gov/data

Response: Thanks for the suggestions. In this study, we intended to use a general urban land use classification for the SSN modeling to highlight the general impact of urbanization on water quality. While more detailed categorization of the land use will certainly provide more detailed investigation, due to sampling cost, the sample size of the current study is not particularly large. To avoid losing too many model degrees of freedom, we have refrained from using more detailed land use land cover data. We present the clarification in the revision as well. We obtained the land use land cover from the New Jersey Department of Environmental Protection, Bureau of GIS, which were products from the NJ’s own mapping effort using color infrared imagery. We hope the revision and clarification will gain your support.

Line 182. How much wastewater is being discharged and where are the plants located. This information should be available from NPDES permits and compliance monitoring reports. Could this impact your results?

Response: We’ve tried including discharge data from the New Jersey Pollutant Discharge Elimination System (NJPDES) but did not identify any specific correlations between E. coli concentrations and number of NJPDES sites upstream of sampling locations. Therefore, this information was not included in the paper. The information about NJPDES was thus deleted in the method section.

Section 2.2:  It would be good to include a table that lists all the input variables/data used and the source/method by which they were obtained. I understand they are listed in the text but summarizing them in a table would be helpful.

Response: Following your suggestion, we added Table 1 to detail the variable/data and the source/method of how they were obtained.

Lines 227-227. This sentence is awkward – please reword. “The response variable were geometric mean concentrations of E. coli for all sampling dates and during storm events, respectively.”

Response: Thank you for your meticulous review. We have revised the sentence to “Two models were established and calibrated. The first model uses the geometric mean concentrations of E. coli for all sampling events as the response variable; the second model uses the geometric mean concentrations of E. coli during storm events only as the response variable.” We hope this modification makes the narrative clearer.

Section 3.1. How many events were considered storm vs. non-storm events? I suggest including the number of events on the boxplot.

Response: Thank you for your suggestion. We have included this specific information in the text but not in the boxplot because most of the sites had the same number of storm and non-storm events.

Line 282-284: Please reword this sentence as it is confusing. It leads the reader to believe that three models were used considering no spatial correlation.

Response: Thank you for your suggestion. We have revised the sentence.

Line 300: Change “wetland use” to “wetland land use”

Response: The term has been revised.

Figure 4: “Size of the circles indicate the standard errors of the predictions, with larger ones showing smaller standard errors.”  Is this correct? Why use larger markers for smaller standard errors?

Response: The maps were created by a default ssn package in R, with larger circles indicating smaller standard errors. However, for better presentation and to accommodate the following suggestions, all data were exported to ArcGIS Pro and new maps were created. Standard errors of the predictions were not shown for simplicity.

Maybe you need to change the size range of the circles. It is difficult to see any variation in size at this scale.

Response: We have further modified the map to make the size differences more distinguishable.

I also suggest using a different color scheme - the stream layer tends to blend in with the points, especially for fig 4a.

Response: We have modified the maps following your suggestions. Thank you.

I don’t think the x and y coordinate information is important here. I suggest a background of a watershed map.

Response: Thank you for your suggestion. We have modified the maps following your suggestion.

Section 3.3: It would be interesting to see a plot of observed vs. predicted E. coli with a 1:1 line. This would illustrate the model fit across the range of observations – could be color coded as storm/non-storm.

Response: Thank you for your suggestion. While we do agree a plot of the observed vs. predict E. coli would be illustrative, since we have included the modeling results and relevant performance statistics in Table 2, plotting the predict and observed points might not add more to the discussion than is current present other than the visual appealing. We decided to stick with the narrative based on the Table. We hope the decision of not adding the plot will gain your support and understanding.

I suggest referring to “study sites” as “sampling locations.” It is just one study site (the Musconetcong River watershed).

Response: Thank you, we have revised it throughout the manuscript.

Section 4.3. Were the rainfall events observed during the sampling period considered extreme events? I do not see any information provided about the storm depth/intensity.

Response: We have added information about storm intensity in Section 3.1.

Line 454: I strongly disagree with this statement “Since field water samplings are time-consuming and costly, water quality modeling can serve as a cost-effective and timely alternative.” Models should only be used in conjunction with sampling programs to enhance our understanding. I would not suggest modeling as an alternative to sampling.

Response: We agree with your suggestion here that models should only be used in conjunction with field sampling to improve water quality monitoring programs. The original text has been modified to avoid misunderstandings.  

Line 458: Again, characterizing modeling as an “alternative” to sampling is very problematic. It can be used to enhance the information we can gain from sampling but never as an alternative.

Response: We have revised it to be “approach.”

Line 477-480: Again, I would not characterize probabilistic modeling as an alternative to sampling. It can increase the information we are able to gain from sampling, but it is NOT an alternative.

Response: We have revised it to be approach, not alternative. Thank you for your meticulous review.

Line 489: “at least 10 – 50 observations” This is confusing. It is 10 or 50. That is a very large range. Ten samples seem very low to establish a robust spatial autocovariance model.

Response: We cited the exact wording from Money and colleague’s text here. We think the last sentence “depending on the watershed size” is the key for this somehow vague statement for the minimum required data points. We hope adding the verbatim citation could slightly clarify the requirement.

Section 4.4. I think you need to add some discussion about how increased sampling would allow for model validation. You do not include model validation in this study, but increased sampling would allow for this. Ideally, you would have had a dataset large enough to separate into training and validation datasets for your model.

Response: Thanks for the suggestions about model validation. We have included this clarification in the section.

Line 521: “all and during storm events.” This wording is awkward, please revise.

Response: we have revised the sentence.

Line 526: Change “wetland use” to “wetland land use”

Response: we have revised accordingly. Thank you again for your meticulous review. This is of great help.

Reviewer 2 Report

General comments:

In this study, spatial stream network (SSN) models were applied to associate key land use variables with elevated concentrations of E. coli in the suburban, mixed-use Musconetcong River Watershed in response to upstream land use attributes (urban, pasture, forest, and wetland) at 11 sampling sites from May to October 2018. Spatial stream network (SSN) modeling provides an empirical approach for urban and suburban water quality stakeholders to analyze the spatial distribution of parameters without first understanding the underlying hydrologic and biogeochemical processes that lead to water quality impairment. In addition, the model allows the identification of potential water pollution "hot spots" and thus the selection of sites for further water monitoring.

E. coli and faecal coliform bacteria are frequently mentioned in the article. It would be good to briefly explain their importance in water quality monitoring.

I believe that the article makes an important contribution to the field of water quality monitoring.

It is written clearly and comprehensively.

After minor changes, I believe it is suitable for publication.

Specific comments

Line 50-51      Water quality criteria were established based on epidemiological studies to investigate rates of gastrointestinal illness among swimmers.

                        I believe that this sentence should be revised, given that the goal of establishing water quality criteria is primarily the protection of human health.

Line 229-230  The rainfall threshold for a storm event was defined as 12.7 mm (0.5 inch) within 36 hours.

                        Could you please explain how rainfall threshold was determined?

Line 401-404  In fact, more than 400,00 cases of acute gastrointestinal illness (AGI) were attributed to a drinking water treatment plan overwhelmed by high turbidity load after a period of heavy precipitation in Milwaukee, Wisconsin in 1993 [61]

Do you mean more than 400,000 cases?

Do you mean “drinking water treatment plant”

Author Response

General comments:

In this study, spatial stream network (SSN) models were applied to associate key land use variables with elevated concentrations of E. coli in the suburban, mixed-use Musconetcong River Watershed in response to upstream land use attributes (urban, pasture, forest, and wetland) at 11 sampling sites from May to October 2018. Spatial stream network (SSN) modeling provides an empirical approach for urban and suburban water quality stakeholders to analyze the spatial distribution of parameters without first understanding the underlying hydrologic and biogeochemical processes that lead to water quality impairment. In addition, the model allows the identification of potential water pollution "hot spots" and thus the selection of sites for further water monitoring.

Response: Thank you for your meticulous review. We appreciate your comments which make the manuscript clearer and better.

  1. coli and faecal coliform bacteria are frequently mentioned in the article. It would be good to briefly explain their importance in water quality monitoring.

Response: Thank you for this clarifying comment. E. coli and fecal coliform are both fecal indicator bacteria. We have added briefly their importance in water quality monitoring in the first paragraph of the Introduction.

I believe that the article makes an important contribution to the field of water quality monitoring.

Response: we appreciate your making the manuscript better.

It is written clearly and comprehensively.

Response: We appreciate your making the manuscript better.

After minor changes, I believe it is suitable for publication.

Specific comments

Line 50-51      Water quality criteria were established based on epidemiological studies to investigate rates of gastrointestinal illness among swimmers.

                        I believe that this sentence should be revised, given that the goal of establishing water quality criteria is primarily the protection of human health.

Response: Thank you for this comments. We have revised this sentence to “Water quality criteria were hence established to protect public health based on epidemiological studies that investigate the relationships between the rates of gastrointestinal illness among swimmers and the levels of FIBs.” We hope this clarification gains our support.

Line 229-230  The rainfall threshold for a storm event was defined as 12.7 mm (0.5 inch) within 36 hours.

                        Could you please explain how rainfall threshold was determined?

Response: The rainfall threshold for a storm event was defined by the Rutgers Cooperative Extension Water Resources Program (reference #37) in the Watershed Restoration and protection Plan for the Musconetcong River Watershed. We have added the explanation and the reference to make the source clear. We hope the modification and addition would gain your support.

Line 401-404  In fact, more than 400,00 cases of acute gastrointestinal illness (AGI) were attributed to a drinking water treatment plan overwhelmed by high turbidity load after a period of heavy precipitation in Milwaukee, Wisconsin in 1993 [61]

Do you mean more than 400,000 cases?

Response: Thank you for catching this typo. Yes, it is supposed to be 400,000 cases. We have corrected this number.

Do you mean “drinking water treatment plant”

Response: Thank you for catching this typo. Yes, it is supposed to be drinking water treatment plant. We have corrected this typo.

Reviewer 3 Report

The manuscript titled “Predicting fecal indicator bacteria using spatial stream network models in a mixed land use suburban watershed in New Jersey, USA” is an interesting approach for fecal bacteria modeling. In this study, spatial stream network (SSN) models were used to associate land use variables with E. coli in a suburban watershed in New Jersey. Some remarks are given to improve this manuscript:

In the introduction section, some advantages and inconveniences are provided for mechanistic and empirical models. However, the discussion about the drawbacks of stochastic models is limited. This situation must be discussed in detail. Empirical models require a large amount of data. A high expertise knowledge is also required to identify associations between response variables and external factors, such as the use of multivariate or non-parametric statistics.

In section 2.1. It is not clear if water samples were collected during storm events. In line 230, it is mentioned that geometric mean concentrations of E.coli were calculated during storm events. The authors must provide the hydrological behavior in the study area to identify the storm events periods.

In line 231. How did the authors define 12.7 mm as the rainfall threshold? Please, describe the procedure for this determination.

In lines 216-227, the authors describe the methodology to perform the SSN modeling. To be replicable, the results of every stage described in these lines must be presented in the results section (landscape network (LSN), contributing areas, RCA attributes and area, watershed attributes, upstream distance, and segment proportion influence, among others).

Does the alpha value mentioned in line 244 correspond to the significance level? If so, this means that a statistical analysis was carried out, but this is not mentioned in the manuscript. The authors only mentioned some error measures, such as RMSPE, LOOCV, and R2, which were used to describe the predictive capability among models. The authors should better explain this situation.

I suggested that the maps show the concentrations of E. coli in their original units. Currently, these maps show the log10 geometric mean concentrations of E. coli. In this sense, this manuscript could also discuss the predictive capability of the models using the original values of E.coli. Are these models also limited by the normal behavior of the response variable? Please, include a discussion in the manuscript about this situation.

Lines 269-271. “During storm events (threshold 269 defined as 12.7 mm within 36 hours), the geometric mean concentrations of E. coli among 270 the mainstem sites ranged from 303 to 811 CFU/100 mL), while those among the tributary 271 also had a greater variation (585.7 to 7300 CFU/100 mL)” Were these concentrations measured in the field?

Figure 3 shows the % of land use at different mainstem sites and sampling locations. The authors must describe in detail how these results were obtained. This is not mentioned in the materials and methods section.

Figure 4 is not clear. Please, modify this figure. A better figure can be provided to illustrate the observed and predicted log 10 geometric means concentrations of E. coli.

Lines 390 – 392. This study does not demonstrate a significant association between urban land use with log10 geometric concentration. There is no statistical test to validate this situation. Please, rewrite this sentence.

Lines 399-401. “The significant 399 correlation was only found between…” Was a Pearson correlation carried out to demonstrate this sentence? Please rewrite this sentence.

Line 404. “A strong negative association (although insignificantly) was identified…” What was the criterion used to consider a strong association?  Is it strong but not significant? What does it mean? Please rewrite this section.

Lines 520-521 and lines 522-523. These sentences must be rewritten since no statistical correlation analysis was carried out.

Please, correct several typos found throughout the manuscript.

Author Response

The manuscript titled “Predicting fecal indicator bacteria using spatial stream network models in a mixed land use suburban watershed in New Jersey, USA” is an interesting approach for fecal bacteria modeling. In this study, spatial stream network (SSN) models were used to associate land use variables with E. coli in a suburban watershed in New Jersey. Some remarks are given to improve this manuscript:

In the introduction section, some advantages and inconveniences are provided for mechanistic and empirical models. However, the discussion about the drawbacks of stochastic models is limited. This situation must be discussed in detail. Empirical models require a large amount of data. A high expertise knowledge is also required to identify associations between response variables and external factors, such as the use of multivariate or non-parametric statistics.

Response: Following your suggestion, we have added some discussion regarding the limitations about regression models in the introduction section.

In section 2.1. It is not clear if water samples were collected during storm events. In line 230, it is mentioned that geometric mean concentrations of E.coli were calculated during storm events. The authors must provide the hydrological behavior in the study area to identify the storm events periods.

Response: We have collected field samples and brought them to lab to determine E. coli counts. We then checked with weather data and compare with the threshold to decide if the particular sample collected at certain day could be treated as storm event samples afterwards. We have added this specific detail to section 2.1.

In line 231. How did the authors define 12.7 mm as the rainfall threshold? Please, describe the procedure for this determination.

Response: The current study situated within the context a state-wide water quality monitoring and evaluation study, we used the same threshold (0.5 inches within 36 hours of collection) defined in the state-wide study (done by the Rutgers Cooperative Extension Water Resources Program). We have added the reference in this revision and clarified the definition of storm event. We hope this revision will address your concern.

In lines 216-227, the authors describe the methodology to perform the SSN modeling. To be replicable, the results of every stage described in these lines must be presented in the results section (landscape network (LSN), contributing areas, RCA attributes and area, watershed attributes, upstream distance, and segment proportion influence, among others).

Response: Thank you for your meticulous review. Before we export the dataset to R for SSN modeling, all the steps mentioned above were conducted in ArcGIS using STARS toolset to create a landscape network (LSN). The entire process is automated, and since there are 427 nodes in the stream network, it would not be feasible to show all 427 pairs of data after each steps. We did cite the STARS toolset in ArcGIS to facilitate the understanding of the data exploration and preparation.

Does the alpha value mentioned in line 244 correspond to the significance level? If so, this means that a statistical analysis was carried out, but this is not mentioned in the manuscript. The authors only mentioned some error measures, such as RMSPE, LOOCV, and R2, which were used to describe the predictive capability among models. The authors should better explain this situation.

Response: The alpha value of 0.05 was used to indicate whether a watershed variable (urban, agricultural, forest, or wetland) has significant correlations with E. coli concentrations in the linear and SSN models. The sentence has been revised for clarifications.

I suggested that the maps show the concentrations of E. coli in their original units. Currently, these maps show the log10 geometric mean concentrations of E. coli. In this sense, this manuscript could also discuss the predictive capability of the models using the original values of E.coli. Are these models also limited by the normal behavior of the response variable? Please, include a discussion in the manuscript about this situation.

Response: Thank you for your suggestion. The original values of E. coli didn’t not pass the normality test (Shapiro-Wilk test). Therefore, we log-transformed the E. coli values to follow the normal distributions. This information has been added in Section 2.2 and Section 3.2 for clarifications.

Lines 269-271. “During storm events (threshold 269 defined as 12.7 mm within 36 hours), the geometric mean concentrations of E. coli among 270 the mainstem sites ranged from 303 to 811 CFU/100 mL), while those among the tributary 271 also had a greater variation (585.7 to 7300 CFU/100 mL)” Were these concentrations measured in the field?

Response: E. coli enumerations were done in a lab following an EPA approved method. We have added the clarification in the revision to avoid confusion.

Figure 3 shows the % of land use at different mainstem sites and sampling locations. The authors must describe in detail how these results were obtained. This is not mentioned in the materials and methods section.

Response: Thank you for your meticulous comment. We clarified that this information is mentioned in Section 2.2: “The percentage of land use was derived from accumulating total area of upstream land use divided by total upstream catchment area processed and generated via the STARS toolset.”

Figure 4 is not clear. Please, modify this figure. A better figure can be provided to illustrate the observed and predicted log 10 geometric means concentrations of E. coli.

Response: Thank you for this comment. The maps were created by the ssn package in R. However, for better presentation, all data were exported to ArcGIS Pro and new maps were created in this revision.

Lines 390 – 392. This study does not demonstrate a significant association between urban land use with log10 geometric concentration. There is no statistical test to validate this situation. Please, rewrite this sentence.

Response: In table 2, we have demonstrated that urban land is significantly related with the log10 geometric mean concentrations of E. coli (at 95% confidence level). We have also discussed the significant association between upstream urban land use and the geometric mean concentration of E. coli in section 3.2. We hope the clarification would gain your support.

Lines 399-401. “The significant 399 correlation was only found between…” Was a Pearson correlation carried out to demonstrate this sentence? Please rewrite this sentence.

Response: Thank you for pointing this out. We have clarified that the significant relationship between concentration of E. coli and the pastureland use in the revision in section 4.2, as presented in Table 2.

Line 404. “A strong negative association (although insignificantly) was identified…” What was the criterion used to consider a strong association?  Is it strong but not significant? What does it mean? Please rewrite this section.

Response: Thank you for catching this oversight. The word “strong” has been removed to avoid confusion, and the statistical significance is considered marginally significant.

Lines 520-521 and lines 522-523. These sentences must be rewritten since no statistical correlation analysis was carried out.

Response: The SSN model is a spatial regression model in essence that can estimate the relationship between stream variables. The relationships revealed by the SSN model, as revealed in Table 2 is the basis for these statement/sentences. We clarified that the statistical analysis was performed as part of SSN modeling. We hope the clarification will help readers better understand the analysis.

Please, correct several typos found throughout the manuscript.

Response: Thank you for your meticulous review. We have gone through the manuscript carefully to correct any language errors.

Thank you again for your insightful comments. This really helps a lot for improve our maunscript.

Round 2

Reviewer 3 Report

No further remarks are given.

Author Response

Dear reviewer:

Thank you for your remarks regarding the research design and method's description. We have further enriched the research design and method section (specifically sections 2.2 and 2.3). We hope the addition and enrichment would gain your support. Thank you for your review and comments.